Corrected: Publisher correction

# Long-term functional and structural preservation of precision-cut human myocardium under continuous electromechanical stimulation in vitro

Carola Fischer[1], Hendrik Milting[2], Evelyn Fein[1], Elisabeth Reiser[1], Kun Lu[1], Thomas Seidel[3], Camilla Schinner[4], Thomas Schwarzmayr[5], Rene Schramm[2,6], Roland Tomasi[1,7], Britta Husse[1], Xiaochun Cao-Ehlker[1], Ulrich Pohl[1,8] & Andreas Dendorfer[1,8]

In vitro models incorporating the complexity and function of adult human tissues are highly desired for translational research. Whilst vital slices of human myocardium approach these demands, their rapid degeneration in tissue culture precludes long-term experimentation. Here, we report preservation of structure and performance of human myocardium under conditions of physiological preload, compliance, and continuous excitation. In biomimetic culture, tissue slices prepared from explanted failing human hearts attain a stable state of contractility that can be monitored for up to 4 months or 2000000 beats in vitro. Cultured myocardium undergoes particular alterations in biomechanics, structure, and mRNA expression. The suitability of the model for drug safety evaluation is exemplified by repeated assessment of refractory period that permits sensitive analysis of repolarization impairment induced by the multimodal hERG-inhibitor pentamidine. Biomimetic tissue culture will provide new opportunities to study drug targets, gene functions, and cellular plasticity in adult human myocardium.

[1] Walter-Brendel-Centre of Experimental Medicine, University Hospital, LMU Munich, Marchioninistr. 27, 81377 Munich, Germany. [2] Erich & Hanna Klessmann Institute, Clinic for Thoracic and Cardiovascular Surgery, Heart and Diabetes Center NRW, Ruhr-University Bochum, Georgstr. 11, 32545 Bad Oeynhausen, Germany. [3] Institute of Cellular and Molecular Physiology, Friedrich-Alexander University Erlangen-Nürnberg, Waldstr. 8, 91054 Erlangen, Germany. [4] Institute of Anatomy, Ludwig-Maximilians-University Munich, Pettenkoferstr. 11, 80336 Munich, Germany. [5] Institute of Human Genetics, Helmholtz Zentrum München, Ingolstädter Landstr. 1, 85764 Neuherberg, Germany. [6] Clinic of Cardiac Surgery, University Hospital, LMU Munich, Marchioninistr. 15, 81377 Munich, Germany. [7] Clinic of Anaesthesiology, University Hospital, LMU Munich, Marchioninistr. 15, 81377 Munich, Germany. [8] German Center for Cardiovascular Research, partner site Munich Heart Alliance, Munich 80802, Germany. Correspondence and requests for materials should be addressed to A.D. (email: andreas.dendorfer@med.uni-muenchen.de)

Efficient translation of basic biomedical research requires information on the impact of molecular targets in systems with increasing complexity, progressing from cultured cells to isolated organs and living subjects. Prior to the involvement of in vivo studies, experimental systems would be desired that provide intact, multicellular, three-dimensional tissue composition in combination with free choice of external manipulation, thus enabling disease modelling and therapy simulation. Such in vitro systems may include cells and tissues of human origin, a feature that is important in cardiac research since species-related peculiarities of mechanical load, heart rate and electrophysiology impede interpretation of findings obtained, e.g., in mouse models[1,2]. Unfortunately, current experimental models of human myocardium involve severe limitations. Viability and function of trabecular muscle strips and of isolated cardiomyocytes decline rapidly[3], whilst the artificial myocardial tissue composed of differentiated stem cells still represents an immature phenotype[4,5], even though cardiomyocyte maturation can be greatly improved by advanced biomimetic stimulation[6].

With the aim to facilitate the use of adult myocardium, methods have been developed for the automated preparation of viable tissue slices from animal and human hearts[7–9]. Such complex tissue models have been proposed for disease and drug research[10,11], but the limited stability has restricted their use to short-term experiments. Initial attempts of organotypic cultivation utilized the adherence on a filter surface to stabilize tissue dimensions, thus achieving survival of human myocardial slices for up to 4 weeks[7]. While this technology indicated the requirement of mechanical forces for the maintenance of cultivated myocardium, it did neither permit excitation and active contraction of the tissue, nor could it prevent the rapid decline of contractile force in vitro. In view of the essential impact of excitation and contraction for the maintenance and development of myocardial differentiation[12–14], we hypothesized that close simulation of biomechanical and electrical conditions might be adequate for resolving the limitations of unloaded tissue culture.

Bioreactors for the cultivation of muscle tissue have been constructed in many variations[12,15,16]. Most of them have been built with the intention of promoting differentiation or maturation of stem cell-derived cardiomyocytes, and rely on developmental properties of those tissues, such as spontaneous excitation, substrate adhesion or adaptive growth. These requirements, however, are not met by adult myocardial tissue slices. Multifactorial incubation systems providing mechanical load, excitation and oxygen supply to adult myocardial tissue preparations have been shown to support viability and force development for up to 6 days, but these systems were based on technically demanding incubators and permitted maintenance of few samples at a time[13,17]. With the aim of processing and culturing a multitude of tissue specimens, we set out to provide physiological conditions in exchangeable incubation chambers that should permit parallel, easily expandable operation within a standard $CO_2$ incubator.

Human myocardial tissue slices may be suitable for many research applications. Investigations of contractility, electrophysiology, excitation conduction and receptor identification have been performed in the past[7,11,18]. Typically, such assessments can be made only once for each tissue sample so that the high variability of human material is likely to obscure treatment effects. A tissue culture model may increase the sensitivity of such studies by permitting observation of temporal developments within the same experiment. This requires treatment responses to be determined repeatedly under unimpaired culture conditions. To demonstrate the benefits of this approach, we tested the significance of chronically cultured myocardium for the assessment of pro-arrhythmic drug effects under the particular condition that

such effects occur in a greatly delayed manner or result from drug interactions.

The presented technology of biomimetic tissue culture simulates mechanical and nutritional conditions of natural myocardium to the extent that viability and function of easily prepared human myocardial tissue slices can be maintained over weeks. Experiments based on cultured myocardium offer the opportunity to study chronic interventions and slowly accumulating effects. These capabilities enable more reliable drug safety evaluation and provide a basis for more realistic disease modelling and therapy simulation.

## Results

**Simulation of myocardial biomechanics and excitation.** With the aim to fulfil biomechanical, electrical and metabolic requirements for the cultivation of adult myocardium, we designed biomimetic cultivation chambers (BMCCs), as well as an electronic stimulation and recording device (Fig. 1). For mounting into BMCCs, heart muscle slices were glued to thin plastic triangles with parallel orientation of muscle fibres in between the fixation. Free tilting allowed the tissue-holding triangles to adapt to uneven force distribution or non-aligned fibre orientation (Fig. 1d). A linear relationship between shortening and contraction force was implemented by fixation of one end of the muscle to a steel spring wire whose elastic constant (75 mN mm$^{-1}$) was chosen so as to generate normal systolic left ventricular wall tension (15 kN m$^{-2}$)[19] at about 6% shortening of a typical myocardial specimen of $5 \times 5 \times 0.3$ mm$^3$ dimensions. The other end of the tissue was hooked to a linear drive that was manually adjusted to accommodate tissue dimensions and to generate a diastolic preload of 1 mN, corresponding to a normal mean diastolic wall stress of 0.66 kN m$^{-2}$ (Fig. 1a–d)[19]. Force of muscle contraction was continuously derived from the displacement of the magnetic tip of the spring wire. Its magnetic field was detected by a 3D-magnetic field sensor that was placed at a distance generating a maximum of 1.2 mT magnetic flux at right angle to the sensor surface. Magnetic flux responded to displacement of the muscle fixation point with a sensitivity of 7 T m$^{-1}$. The signal/noise ratio of the sensor permitted determination of muscle contraction with a spatial resolution of 0.5 μm.

Electrical stimulation was implemented with an adjustable current source that generated bipolar impulses (typically 1 ms charging and discharging currents separated by a 1 ms interval), and zero current in between. Constant current stimulation provided the advantage of not being influenced by electrode polarization or spurious resistances. By exact balancing of electrode charge, electrochemical reactions were reduced to the extent that unchanged electrodes could be stably operated over months. Typically, tissue excitation was achieved at pulse currents of 20–50 mA, corresponding to an electric field strength of 0.15–0.4 V mm$^{-1}$. When demanded by the tissue excitation threshold, stimulation intensity was increased to a maximum of 75 mA and 3 ms pulse duration.

**Cold preservation of explanted failing myocardium.** To optimize the availability of explanted human tissue, surgical specimens were collected at the local (Munich) as well as a remote (Bad Oeynhausen) clinic. In the latter case, samples were sent on ice by a standard courier service which imposed a delay of up to 32 h to the preparation of tissue slices. Maximum tolerance to hypothermic preservation was tested by intentional prolongation of this condition to up to 72 h. Slices prepared from tissue specimen after up to 55 h cold storage showed no impairment of maximum twitch force, as compared with immediately processed samples, whereas longer cold exposure nearly abolished

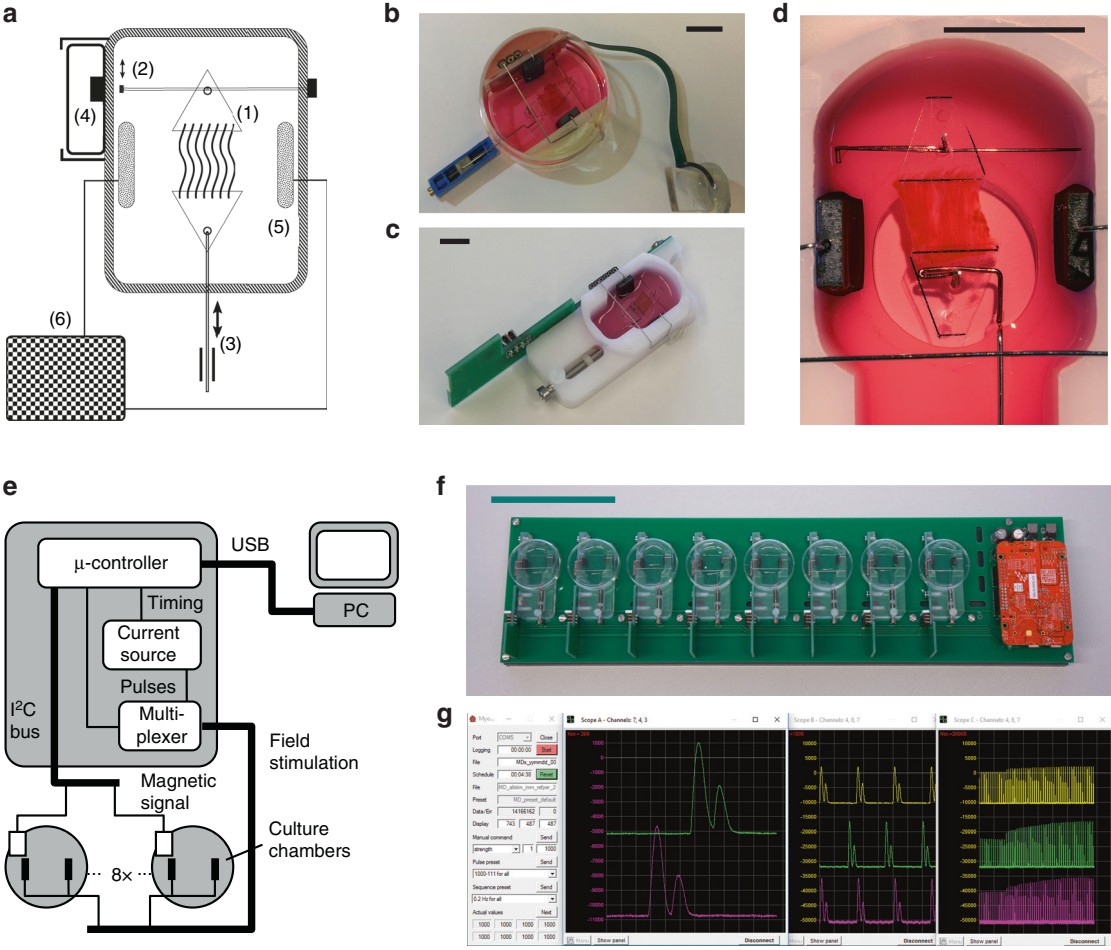

**Fig. 1** Biomimetic culture system. **a** Schematics of the culture chamber depict attachment of tissue slice via plastic triangles (1) to spring wire with magnetic tip (2) and to linear drive (3). Changes in magnetic field are detected by an integrated sensor (4). Field stimulation is provided by graphite electrodes (5) connected to a constant current pulse generator (6). **b-c** Implementations based on standard dish and machined POM. **d** Detailed view of tissue slice fixation and stimulation electrodes. **e** Schematic of controller electronics. Signals from the magnetic field sensors are processed by a microcontroller that also provides bipolar stimulation pulses via a multiplexed current source. **f** Control unit and eight biomimetic culture chambers (BMCCs) are combined on a circuit board that can be operated in a standard $CO_2$ incubator. **g** Recording software. Data are transmitted via USB to an external PC for recording and scheduled execution of stimulation protocols. Black bars represent 10 mm, the green bar 100 mm distances

contractility (Fig. 2a). An initial and reversible increase in tonic contracture indicated incipient injury in tissues cooled for more than 32 h. After standard transport (< 32 h), contracture occurred only accidentally and was not associated with cooling duration. Further characterization of transported myocardial samples also revealed a distinct response to isoprenaline, extensive influence of stretch on active and passive force development, and a definitively positive force/frequency relationship (Fig. 3a–d). These tissues were considered adequate for long-term cultivation.

**Development of contractility during biomimetic culture.** Myocardial samples of 18 patients were used for biomimetic culture experiments. Patient characteristics are listed in Supplementary Table 1. Up to 30 vital slices could be prepared from each tissue specimen. After insertion of slices into BMCCs, we occasionally observed spontaneous contractions, which ceased within 1 h of equilibration. All slices responded with regular contractions upon initiation of electrical stimulation. Individual slices producing less than 30% of the average contractility were replaced by new ones. This measure of standardization was applied to 10–20% of the slices. Regular contractions were induced by continuous pacing (0.2 Hz), and were monitored for up to 4 months (Fig. 2c, d). Slices obtained from four tissue

specimens underwent premature deterioration within 2 weeks of culture, as evidenced by a decline of contractility, tonic contracture or increase in excitation threshold. They were excluded from the experiments. The targeted culture duration of 4 weeks was surpassed in 12 of 18 preparations. Slice cultures could be generated from tissues with various medical histories of the donors. However, all three cases of heart failure with infarction or ischaemia diagnosed as the pathogenic factor yielded tissue samples with impaired culture stability (Supplementary Table 1). The typical time course of force development during biomimetic culture attained a minimum after 6–24 h of culture which was followed by slow recovery to initial levels during the following 3–4 weeks (Fig. 2d). Spontaneous recovery of contractility was anticipated when continuous $\beta_1$-adrenergic stimulation was initiated a the 2nd day of culture (Fig. 2e). In order to test the requirement and efficiency of biomimetic culture conditions, continuous electrical stimulation or medium agitation was suspended for various durations (Fig. 2f). Rapid decline of contractility during 15 min of unstirred culture, and incomplete recovery after prolongation of this condition indicated the requirement of medium agitation for adequate $O_2$ supply. Cessation of electrical stimulation abolished regular contractions, but did not affect excitation and force development after short

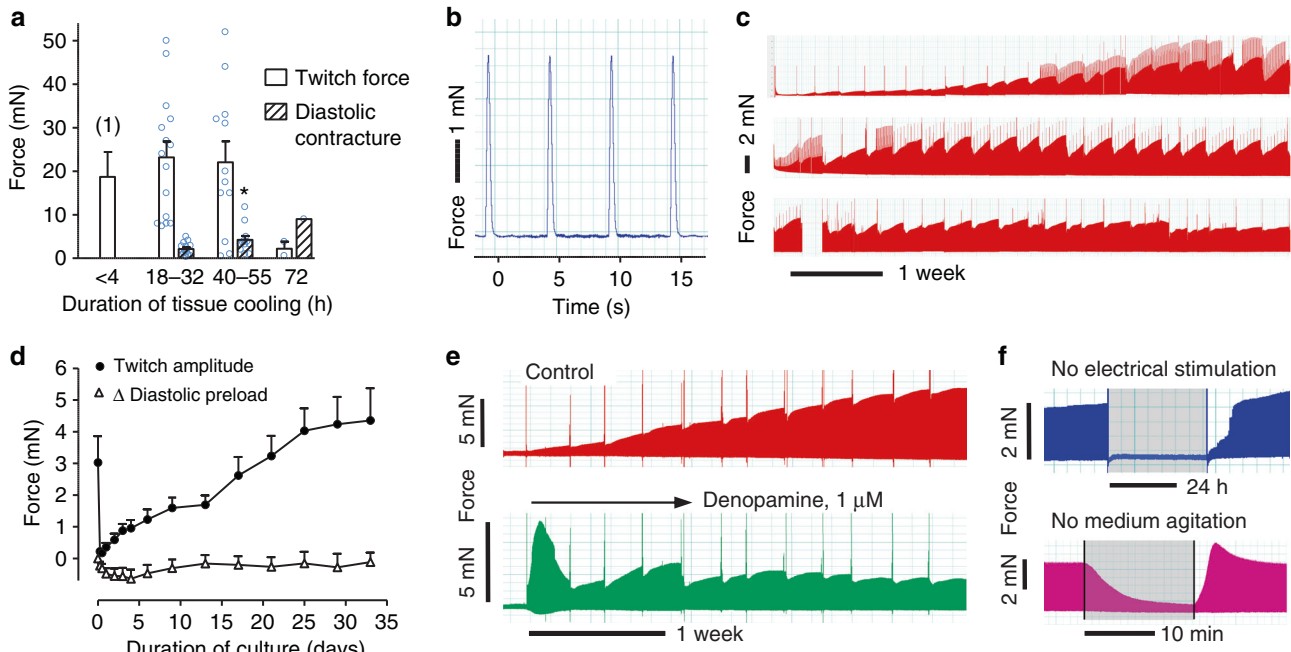

**Fig. 2** Force development of cardiac slices before and during biomimetic culture. **a** Maximum twitch force and diastolic contracture of cardiac slices prepared after various durations of hypothermic tissue storage. Measurements were taken under isometric conditions in an organ bath. Data of immediately processed tissue ((1), $n = 8$) have been assessed in hypertrophic myocardium in a previous study[7]. Tonic contracture increased in slices prepared after prolonged storage (40-55 h, $n = 12$) in comparison to standard transport (18–32 h, $n = 15$, *$P<0.05$, $t$ test). Tissue did not tolerate 72 h of cooling ($n = 2$). Data are displayed as mean ± SEM. **b** High resolution recording of twitch force in BMCC. **c** Continuous contractility recording over 4 months. Periodic breakdowns of contraction force corresponded to medium exchange intervals (36–48 h). Positive spikes of contractility were produced by stimulation protocols employed for the assessment of refractory periods. **d** Time course of twitch amplitude and preload during the initial 5 weeks of cultivation (mean ± SEM of 13 samples taken from nine specimen). **e** Long-term effect on contractility of β₁-receptor stimulation initiated on the 2nd day of culture (denopamine, 1 μM, representative example of five independent experiments). **f** Contractility during omission of electrical stimulation or medium agitation for 30 h and 15 min, respectively (representative examples). Source data of 2a and 2d are provided as Source Data file

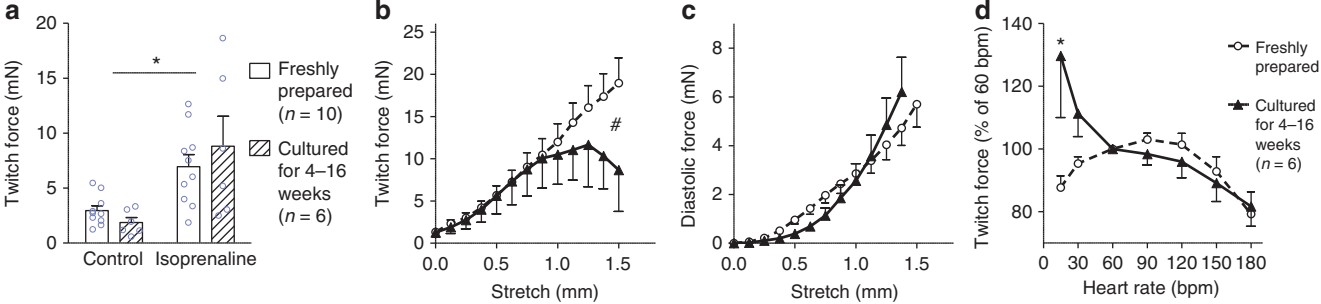

**Fig. 3** Biomechanics of myocardial tissue before and after long-term biomimetic cultivation. **a** Twitch force at 1 mN preload, as determined under isometric conditions in an organ bath in freshly prepared ($n = 10$) and cultured tissues ($n = 6$) before and after application of isoprenaline (1 μM) (*$P<0.05$ control vs. isoprenaline). **b–c** Response of twitch force (**b**) and diastolic force (**c**) to tissue stretch ($^{\#}P<0.05$ fresh vs. cultured, $n = 10$ for fresh tissues, $n = 6$ for cultured tissues). **d** Response of twitch force to variation of beating rate ($n = 9$ for fresh tissues, $n = 6$ for cultured tissues, *$P<0.05$ fresh vs. cultured, bpm = beats per minute). Data are presented as mean ± SEM. Effects were evaluated by two-way ANOVA. Source data are provided as Source Data file

intervals. In contrast, 30 h of inactivity provoked electrical remodelling that suppressed contractility after reinstatement of stimulation, and delayed its recovery for many hours.

Tissue biomechanics were characterized after various durations of biomimetic culture and were compared with those of uncultured tissues. After more than 4 weeks in culture, contractile performance of cultured slices was similar to that of fresh tissue and could be enhanced effectively by isoprenaline (factor 5.5, Fig. 3a). In the physiological range of strain (< 1 mm stretch, corresponding to < 20% distension) fresh and cultured tissues also responded with an equivalent increase in contraction force to mechanical stretch

(Fig. 3b). Cultured tissues displayed a reduced maximum contractility at more intense stretch which was related to the occurrence of arrhythmias, such as delayed contractions and tachycardia. Cultured tissue also developed slightly higher passive forces in response to strain (Fig. 3c), however, this tendency did not reach statistical significance. While cultured and fresh tissues quite similarly declined in contractility when pacing rates increased from 60 to 180 bpm, cultured tissues exclusively responded with a gain in twitch force to bradycardia (Fig. 3d). This reaction might reflect an adaptation of electromechanical coupling to the low beating rate employed in biomimetic culture (0.2 Hz).

**Alterations in tissue structure and gene expression**. Immediately after preparation or after various periods of biomimetic cultivation, myocardial slices were processed for histological examination or mRNA sequencing. Freshly prepared, as well as cultivated for longer than 5 weeks, myocardial tissue showed dense structure of well-aligned myofibrils with preserved cross-striation (α-actinin), and distinct localization of connexin-43 and N-cadherin at intercalated discs (Fig. 4). Culture-related alterations included perinuclear accumulation of lipofuscin and eosinophilic material (HE staining), and some thinning and irregularities of myofibrillar structure. Staining of connective tissue markers vimentin and α-smooth muscle actin (SMA) demonstrated the absence of fibrosis in cultured tissues, but also indicated a slight increment of SMA-positive cells. Accumulation of mesenchymal cells expressing either SMA or vimentin was observed only at the surface of the tissue slices, possibly reflecting a response to cutting injury. The distinct localization of N-cadherin at intercalated discs indicated intact cell-to-cell contacts via adherens junctions, thus providing the basis for force transmission and contractility of the cultured myocardium. Quantification of myocyte structural features demonstrated preservation of sarcomeres and of the transverse tubular system in cultured tissues, as well as the absence of myocyte hypertrophy or atrophy (Supplementary Figs. 1–3).

Messenger RNA was sequenced in myocardial slices from three patients prior to and after 8, 14, 24 and 35 days of culture. Slices analysed at days 14 and 35 were derived from the same patient. Analysis of highly expressed genes and consequential enriched gene ontologies indicated that a general adaptation of gene expression occurred during the first 8 days of culture (Tables 1 and 2, Supplementary Table 2). Prominent changes included components of excitation–contraction coupling, of the extracellular matrix, and targets of adrenoceptor signalling. For important components of cardiomyocyte contractility, these initial alterations regressed, and in some instances (MYH7, TTN) were even outweighed during the following 4 weeks of culture. This regulation may explain the consistent recovery of contraction force during the corresponding period of culture (Fig. 2d). In contrast, initial induction of extracellular matrix proteins and suppression of β-adrenergic targets were sustained throughout the culture period. Known markers of pathologic processes, such as contractile failure, hypertrophy or hypoxia tended to be expressed at lower levels in cultured than in native myocardium (Tables 1 and 2). This trend was either manifest initially or evolved during prolonged culture. Particular growth factors were found to be highly upregulated, and might therefore be involved in the induction of matrix components or in the recovery of excitation–contraction coupling. Most effective alterations in gene expression after the first week of culture pointed to an active role of matrix–integrin interaction and of neurotrophic factors, as well as to an attenuation of inflammation and of WNT-signalling as further mechanisms of remodelling in tissue culture (Tables 1 and 2). Gene attribution to functional ontologies confirmed initial suppression and slow recovery during culture of excitation–contraction coupling in general (Supplementary Table 2). Exceptions were evident for genes negatively associated with heart rate (GO:0001985) whose upregulation was possibly induced by slow pacing. Likewise, the

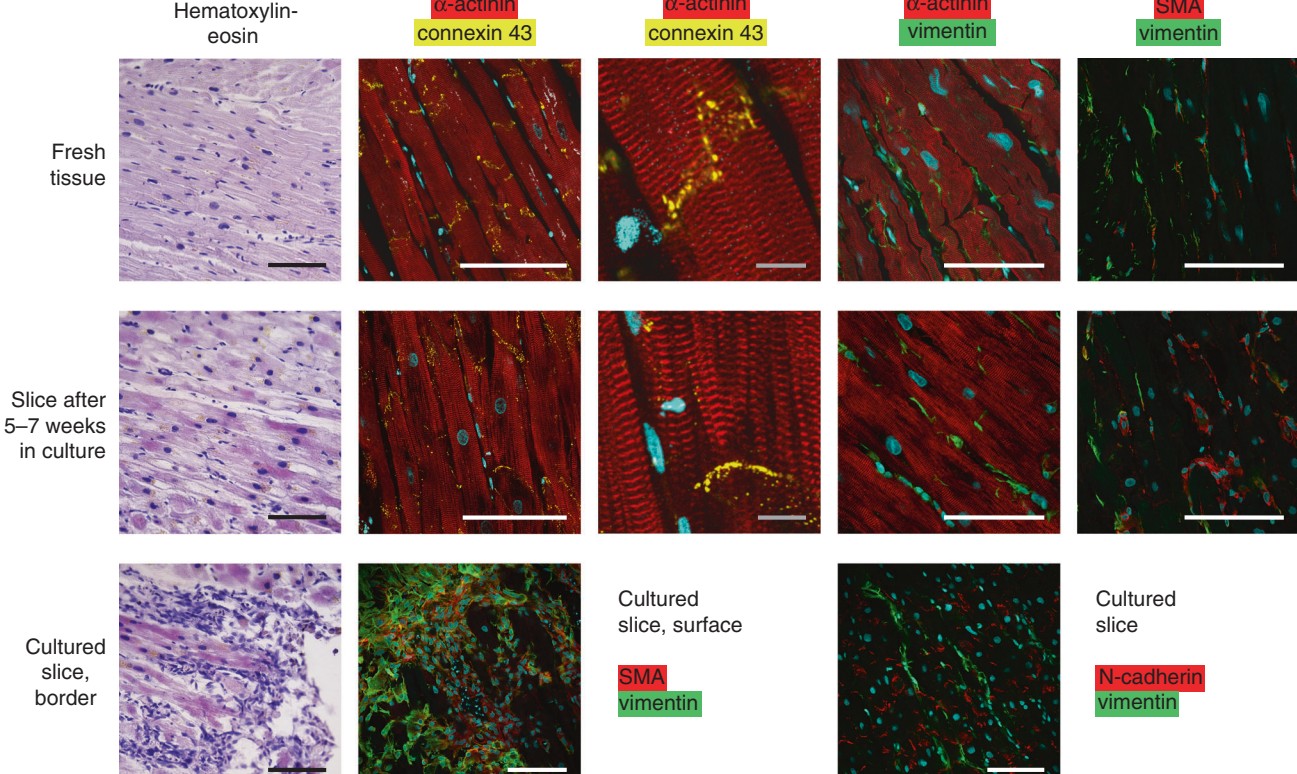

**Fig. 4** Myocardial tissue structure after long-term cultivation. Fresh samples of failing myocardium (1st row) were compared with tissues cultured for five or more weeks (2nd row). Staining of structural cardiomyocyte proteins (α-actinin, connexin-43) and connective tissue markers (vimentin, α-smooth muscle actin (SMA)) are indicated in column headings. Accumulation of mesenchymal cells expressing either SMA or vimentin at the cut surfaces of cultured tissues and distinct localization of N-cadherin at intercalated discs are demonstrated in the 3rd row. DNA is depicted in blue. White and black bars represent 100 μm, grey bars 10 μm distances. Pictures exemplify three independent stainings

**Table 1 Time course of mRNA expression in cultured myocardium**

| Function and protein | Gene | 8 days | 14 days | 24 days | 35 days | Intercept | Slope x35 |
|---|---|---|---|---|---|---|---|
| *Cardiomyocyte contractility* | | | | | | | |
| Troponin C1 | TNNC1 | −1.76 | −2.76 | −2.26 | −2.17 | −2.16 | −0.13 |
| Troponin I3 | TNNI3 | −3.90 | −4.48 | −3.11 | −3.15 | −4.51 | 1.47 |
| Myosin light chain 2 | MYL2 | −2.36 | −3.93 | −2.63 | −2.26 | −3.29 | 0.86 |
| Ryanodine receptor 2 | RYR2 | −2.57 | −1.63 | −1.89 | −0.73 | −2.85 | 1.98 |
| Myosin heavy chain 7 | MYH7 | −3.76 | −1.52 | −2.64 | 0.60 | −4.46 | 4.54 |
| Actin, alpha, cardiac muscle 1 | ACTC1 | −2.35 | −1.44 | −1.96 | −1.48 | −2.22 | 0.71 |
| SERCA 2 | ATP2A2 | −2.08 | −0.80 | −0.70 | −0.86 | −1.84 | 1.27 |
| Titin | TTN | −0.95 | −1.19 | −0.52 | 0.12 | −1.55 | 1.58 |
| *Mitochondria* | | | | | | | |
| Cytochrome c oxidase subunit 5A | COX5A | −0.97 | −2.41 | −1.18 | −2.25 | −1.19 | −0.89 |
| Cytochrome c oxidase subunit 7C | COX7C | −0.99 | −2.38 | −1.37 | −2.11 | −1.27 | −0.77 |
| ATP synthase, F1 complex | ATP5C1 | −0.91 | −2.24 | −1.33 | −2.12 | −1.12 | −0.92 |
| ATP synthase, F0 complex | ATP5H | −0.89 | −2.30 | −1.04 | −2.19 | −1.08 | −0.91 |
| NDUFA4, complex associated | NDUFA4 | −1.36 | −2.39 | −1.41 | −2.13 | −1.58 | −0.43 |
| Ubiquinol-cytochrome c-binding | UQCRB | −1.36 | −2.58 | −1.51 | −2.35 | −1.59 | −0.63 |
| Adenine nucleotide translocase 1 | SLC25A4 | −2.22 | −2.68 | −2.12 | −2.33 | −2.42 | 0.14 |
| *Markers of heart failure (1)* | | | | | | | |
| Natriuretic peptide B | NPPB | −2.36 | −3.50 | −3.04 | −3.78 | −2.38 | −1.37 |
| Attractin like 1 | ATRNL1 | −0.14 | −1.23 | −0.89 | −1.63 | −0.11 | −1.49 |
| Ribosomal protein S7 | RPS7 | 0.27 | −1.11 | −0.77 | −1.24 | 0.14 | −1.47 |
| Actin, alpha 1, skeletal muscle | ACTA1 | −2.78 | −3.03 | −3.11 | −2.75 | −2.95 | 0.06 |
| Msh homeobox 2 | MSX2 | 0.66 | −1.09 | −0.06 | −0.93 | 0.38 | −1.27 |
| Calsequestrin 1 | CASQ1 | −2.12 | −2.71 | −1.92 | −1.76 | −2.60 | 0.81 |
| X-prolyl aminopeptidase 3 | XPNPEP3 | −0.80 | −1.50 | −0.36 | −1.21 | −0.96 | −0.01 |
| *Markers of hypertrophy (2)* | | | | | | | |
| Ankyrin repeat domain 1 | ANKRD1 | 0.13 | −2.87 | −1.92 | −1.90 | −0.73 | −1.57 |
| Aldolase, fructose-bisphosphate A | ALDOA | −0.96 | −0.48 | −0.99 | −0.94 | −0.71 | −0.23 |
| Crystallin alpha B | CRYAB | −0.31 | −2.24 | −1.41 | −1.97 | −0.68 | −1.38 |
| Cysteine and glycine rich protein 3 | CSRP3 | 1.01 | −1.96 | −1.42 | −2.03 | 0.60 | −2.93 |
| FXYD domain containing ion transport regulator 1 | FXYD1 | −1.69 | −1.42 | −1.24 | −0.92 | −1.86 | 0.94 |
| Phosphatidylethanolamine binding protein 1 | PEBP1 | −1.50 | −1.27 | −0.99 | −1.09 | −1.53 | 0.54 |
| Pyruvate dehydrogenase E1 beta | PDHB | −0.96 | −1.51 | −0.97 | −1.33 | −1.09 | −0.17 |
| *Markers of adrenergic activity (3)* | | | | | | | |
| Natriuretic peptide A | NPPA | −3.31 | −2.88 | −3.18 | −3.02 | −3.20 | 0.17 |
| Actin, alpha 1, skeletal muscle | ACTA1 | −2.78 | −3.03 | −3.11 | −2.75 | −2.95 | 0.06 |
| Lectin serine peptidase 1 | MASP1 | −2.00 | −1.86 | −2.44 | −1.52 | −2.18 | 0.39 |
| Nuclear receptor subfamily 4A1 | NR4A1 | −2.51 | −2.85 | −2.01 | −2.77 | −2.55 | 0.02 |
| Fos proto-oncogene | FOS | −1.46 | −2.30 | −0.70 | −2.51 | −1.38 | −0.63 |
| Inhibitor of DNA binding 1 protein | ID1 | −3.03 | −2.07 | −2.72 | −0.99 | −3.42 | 2.10 |
| Kruppel like factor 4 | KLF4 | −2.05 | −2.02 | −1.97 | −1.70 | −2.19 | 0.43 |

Gene expression reflecting alterations of cardiomyocyte differentiation or myocardial pathophysiology during tissue culture. The ratios of gene transcripts in cultured vs. uncultured tissue from the same patient are expressed as Log2-values. Temporal alterations of gene expression were approximated by linear regression of all culture durations. This calculation estimated the change of expression levels at the start of cultivation (intercept), and the temporal trend during 35 days of culture (slope×35). Genes were selected that represent cardiomyocyte characteristics, or the pathology of myocardial disease, such as (1) heart failure[40], (2) cardiac hypertrophy[41,42], and (3) β-adrenergic stimulation[43,44]. Complete mRNA expression data are provided as Supplementary Data 1

lack of adrenergic tone was reflected by genes transmitting the adrenergic enhancement of contractility and heart rate (GO:0003059, GO:0003065).

**Repeated assessment of hERG-channel (Kv11.1) interference.**
Reduced conductivity of the K+-channel encoded by the ether-a-gogo-related gene (*KCNH2*, hERG-channel) is known to cause repolarization impairment leading to prolongation of the myocardial action potential and refractory period. Refractory period was tested during unmodified culture by combining a regular pacing impulse with a second stimulus set at decreasing intervals (Fig. 5a). The longest pulse interval that failed to induce two distinct contractions was considered to reflect the actual refractory period. Durations of 425±24 ms were determined in tissues cultured for 2–6 weeks. Cumulative application of the ± hERG blocker dofetilide immediately increased the refractory period by up to 160±18 ms ($n = 6$, $EC_{50} = 3$ nM, Fig. 5b). In contrast, interference with hERG-channel expression and membrane

integration by the multimodal drug pentamidine was ineffective on the first day of incubation, but elicited a progressive increase in refractory period over the course of 2 weeks (Fig. 5c). At this time, pentamidine (1 µM) was more effective than dofetilide (100 nM), indicating inhibition of repolarizing channels in addition to hERG (312±65 ms, $n = 6$, $P<0.05$, $t$ test). Validity of refractory period assessment for myocardial repolarization was confirmed by intracellular potential recordings that demonstrated prolongation of the action potential duration ($APD_{90}$) by pentamidine (Fig. 5d, e). Dofetilide greatly enhanced the prolongation of action potential duration after pentamidine pretreatment, and provoked various kinds of delayed depolarizations (Fig. 5f, g). This effect was antagonized by alternative stimulation of K+ conductivity.

## Discussion

We present a technology that allows for culture of beating intact adult human myocardium under constant functional

**Table 2 Time course of mRNA expression in cultured myocardium**

| Function and protein | Gene | 8 days | 14 days | 24 days | 35 days | Intercept | Slope x35 |
|---|---|---|---|---|---|---|---|
| *Growth factors* | | | | | | | |
| Growth differentiation factor 15 | GDF15 | 2.98 | 2.07 | 2.26 | 1.32 | 3.19 | −1.79 |
| Hepatocyte growth factor | HGF | 2.21 | 3.12 | 2.66 | 3.17 | 2.30 | 0.84 |
| Opioid growth factor rec. like 1 | OGFRL1 | 1.16 | 2.20 | 0.91 | 2.00 | 1.33 | 0.41 |
| Platelet derived growth factor receptor beta | PDGFRB | 1.34 | 3.66 | 1.93 | 3.56 | 1.64 | 1.70 |
| Placental growth factor | PGF | 3.06 | 4.20 | 2.42 | 3.65 | 3.38 | −0.08 |
| Transforming growth factor beta 1 | TGFB1 | 1.69 | 2.75 | 1.76 | 2.22 | 2.04 | 0.12 |
| Angiotensin I converting enzyme | ACE | 1.87 | 3.02 | 1.94 | 2.05 | 2.43 | −0.36 |
| *Extracellular matrix* | | | | | | | |
| Collagen type I alpha 1 chain | COL1A1 | 3.16 | 4.04 | 2.90 | 3.86 | 3.31 | 0.32 |
| Collagen type I alpha 2 chain | COL1A2 | 2.42 | 3.19 | 2.20 | 3.36 | 2.40 | 0.68 |
| Collagen type III alpha 1 chain | COL3A1 | 2.25 | 3.30 | 2.50 | 3.52 | 2.26 | 1.09 |
| Vimentin | VIM | 0.96 | 1.70 | 1.25 | 1.21 | 1.27 | 0.02 |
| Laminin subunit gamma 1 | LAMC1 | 0.87 | 1.45 | 0.30 | 1.77 | 0.73 | 0.63 |
| Periostin | POSTN | 2.70 | 3.48 | 1.92 | 3.44 | 2.73 | 0.26 |
| Fibronectin 1 | FN1 | 3.24 | 3.89 | 2.41 | 3.35 | 3.49 | −0.46 |
| *Hypoxia-induced targets* | | | | | | | |
| Vascular endothelial growth factor A | VEGFA | −0.59 | 1.18 | −0.98 | −0.38 | 0.26 | −0.78 |
| Solute carrier family 2 member 1 | SLC2A1 | −1.97 | 1.79 | −1.30 | −0.12 | −0.70 | 0.53 |
| Glyceraldehyde-3-phosphate dehydrogenase | GAPDH | −0.20 | −0.19 | −0.24 | −1.02 | 0.18 | −1.02 |
| Protein kinase, cAMP-activated | PRKACA | −0.33 | −0.78 | −1.11 | −0.62 | −0.50 | −0.37 |
| Phosphoglycerate kinase 1 | PGK1 | −0.85 | 0.18 | −0.80 | −0.94 | −0.24 | −0.63 |
| *Positive trend* | | | | | | | |
| Elastin | ELN | −2.68 | −1.10 | −0.75 | 2.12 | −3.88 | 5.67 |
| Nerve growth factor receptor | NGFR | −1.80 | 2.04 | 0.91 | 3.71 | −2.03 | 5.61 |
| Kelch-like family member 38 | KLHL38 | −3.53 | −0.90 | −1.16 | 1.48 | −4.22 | 5.52 |
| X-prolyl aminopeptidase 2 | XPNPEP2 | −1.77 | 2.22 | 1.05 | 3.65 | −1.87 | 5.45 |
| Integrin subunit beta like 1 | ITGBL1 | −2.75 | 0.02 | −0.20 | 2.16 | −3.30 | 5.38 |
| Neurotrophic receptor tyrosine kinase 2 | NTRK2 | −1.88 | −0.43 | −0.61 | 2.45 | −2.98 | 4.95 |
| Phospholipid phosphatase rel. 4 | LPPR4 | −0.39 | 0.67 | 2.76 | 2.60 | −0.96 | 4.10 |
| *Negative trend* | | | | | | | |
| Prolyl 4-hydroxylase alpha 3 | P4HA3 | 3.61 | 4.06 | 2.55 | 0.40 | 5.26 | −4.51 |
| Interleukin 6 | IL6 | 2.56 | 1.20 | 1.13 | −1.18 | 3.43 | −4.33 |
| TNF receptor superfamily member 12 A | TNFRSF12A | 2.71 | −1.30 | −0.96 | −1.48 | 2.18 | −4.21 |
| Cell division cycle associated 2 | CDCA2 | 1.94 | 0.23 | −0.48 | −1.50 | 2.41 | −4.08 |
| Wnt family member 5 A | WNT5A | 2.90 | 1.19 | 0.20 | −0.21 | 3.20 | −3.77 |
| Protein phosphatase 1 regulatory subunit 16 B | PPP1R16B | 0.21 | −0.31 | −2.56 | −2.36 | 0.91 | −3.75 |
| C-X-C motif chemokine ligand 1 | CXCL1 | 3.15 | 1.10 | 0.25 | 0.02 | 3.26 | −3.68 |

Gene expression reflecting myocardial remodelling during tissue culture. The ratios of gene transcripts in cultured vs. uncultured tissue from the same patient are expressed as Log2-values. Temporal alterations of gene expression were approximated by linear regression of all culture durations. This calculation estimated the change of expression levels at the start of cultivation (intercept), and the temporal trend during 35 days of culture (slope ×35). Genes were selected that represent typical culture-related alterations, such as cell proliferation and hypoxia. Exemplary genes are also shown that underwent most extensive induction or downregulation in the course of biomimetic culture (positive or negative trends). Complete mRNA expression data are provided as Supplementary Data 1

monitoring and demonstrate its application to drug safety evaluation. Preservation of tissue function, structure and differentiation in vitro has been achieved for up to 4 months. We suggest that the unprecedented stability is not an effect of one specific feature of the culture setup, but rather results from optimizing several parameters in combination. Obviously, simulation of tissue mechanics by introduction of directional forces constituted a mandatory improvement because identically prepared myocardial slices underwent extensive functional degeneration under culture conditions devoid of mechanical load[7]. However, mechanical load did not appear to be the only requirement for the culture of myocardial slices, since two recent studies did not report preservation of contractility with its application[17,20]. The only previous example of maintenance of contractile competence in culture has been provided with human trabecular muscle fibres that have been kept for up to 6 days in an elaborately designed incubator with careful application of diastolic tension[13]. In contrast to this approach, our integrated biomimetic system provided elastic (as opposed to isometric) contraction, adequate $O_2$ delivery at atmospheric partial pressure and avoidance of electrochemical reactions by charge-balanced stimulation as improved approximations of normal physiology.

These characteristics may summarize the most important advances of the novel technology. Surprisingly, nutritive demands of the myocardium appeared to be fulfilled by serum-free basal medium with glucose as the sole energy source. The absence of external growth stimuli, and application of mechanical load may also explain the observed stability of mesenchymal cells within the tissues. Indeed, constant stretch has been shown to suppress fibroblast proliferation in myocardial slices[20]. In addition, the multicellular nature of the cultured tissue seemed to support cellular interactions also by release of functionally important endogenous mediators, as can be concluded from the breakdown and recovery of contraction force in between each medium exchange (Fig. 2c).

The most prominent culture-induced alteration in myocardial function emerged as the initial loss and subsequent steady recovery of contraction force during the first weeks of culture (Fig. 2c). The absence of physiological stimuli, catecholamines in particular, seemed to account for most of the initial suppression of contractility (Fig. 2e) that was also paralleled by downregulation of genes involved in excitation–contraction coupling, and in adrenoceptor signalling (Table 1). The phenomenon of spontaneous recovery of contraction force remains to be resolved.

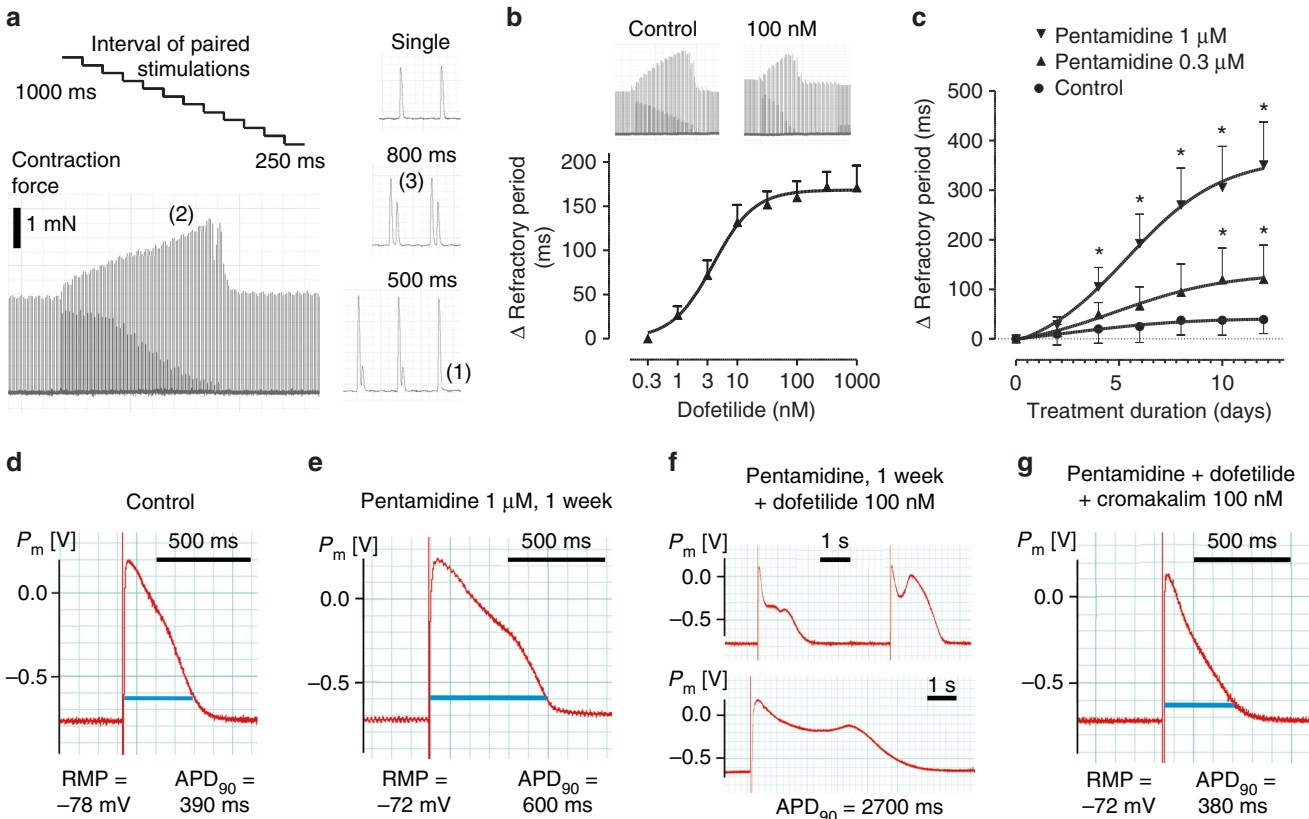

**Fig. 5** Assessment of refractory period and effects of hERG-channel blockade. **a** Stimulation protocol that applied paired stimulation pulses with successively decreasing time-lag. Loss of the subsequent contraction indicated equality with the refractory period (1). Phenomena of post-rest potentiation (2) and attenuation of premature beats (3) were likewise observed. **b** Dose-dependent increase of refractory period by the hERG-channel blocker dofetilide ($n = 5$, mean ± SEM). **c** Delayed increase of refractory period by the multimodal channel blocker pentamidine (*$P < 0.05$ vs. baseline, two-way ANOVA, $n = 5$, mean ± SEM). **d–g** Intracellular recordings of action potentials in tissue slices after 3 weeks biomimetic cultivation either under standard conditions (**d**), or after a 1-week exposure to pentamidine (**e–g**) (representative observations). **f, g** Slices chronically pretreated with pentamidine were exposed acutely to combinations of the hERG blocker dofetilide and the $K_{ATP}$-channel activator cromakalim. $P_m$ = membrane potential, RMP = resting membrane potential, $APD_{90}$ = action potential duration at 90% repolarization. Source data of panels 5b, c are provided as Source Data file

Transcriptome analysis indicated slow upregulation of sarcomeric and $Ca^{2+}$-handling proteins, rather than restoration of β-adrenergic signalling as potential contributions to force recovery. These alterations certainly include adaptation to culture conditions, such as bradycardia (Fig. 3d), but may also imply reversal of the preexistent failing state of the myocardium. The observed overall decline of mRNA expression of markers of heart failure, myocardial hypertrophy and hypoxia supports this hypothesis (Tables 1 and 2). Expression of several growth factors, and of matrix proteins, such as collagens and fibronectin, was found to be induced during culture, but was not accompanied by overt fibrosis or myocyte remodelling (Fig. 4, Supplementary Figs. 1–3). Rather, proliferation of mesenchymal cells appeared to be restricted to the surface of the tissue slices. Notably, we did not find signs of myocyte atrophy, and the transverse tubular system, which in cultured cardiomyocytes deteriorates within 2–4 days[3], was preserved even for several weeks of culture (Supplementary Figs. 1–3).

Biomimetic culture appears to extensively stabilize structure and function of human myocardium in vitro, thereby creating the opportunity to study long-term interventions, genetic manipulation and chronic treatment effects in this therapeutically relevant tissue. The technique also decisively enhances the availability of human tissues by the option to distribute samples with standard courier services. The high tolerance of tissue specimen to cold storage is probably owed to the fact that the model does not depend on reestablishment of perfusion. The automated preparation can be scaled up to a multitude of slices from one specimen, so that the presented technology advances not only temporal but also quantitative possibilities for research on human tissues. For further evolution of this potential, some limitations of the model still need to be addressed. Exposure of tissues to culture conditions clearly induces functional responses that may be desirable, such as adaptation to standardized mechanical load, but may also be artificial, e.g., the loss of adrenergic stimulation and the implementation of bradycardia. On the positive side, the overall adaptation resulted in recovery of contractile force, and may therefore involve mechanisms of reverse tissue remodelling whose identification might be worthwhile. Conversely, pharmacological studies, like the ones presented here, are best performed under stable experimental conditions which require 2–3 weeks of culture to be established. It can be expected that substitution of adrenergic tone will greatly accelerate recovery of contractile performance (Fig. 2e), however, the impact of this intervention needs to be characterized in more detail. Similarly, tissue performance may profit from refinement of the biomechanical parameters of culture. Low-rate stimulation and modest diastolic load were chosen to minimize tissue stress. These conditions have been capable of avoiding tissue hypoxia (Table 2), and to keep sarcomere length of slack tissues constant during culture (Supplementary Fig. 1a). However, bradycardia

emerged as a strong inducer of tissue adaptation with unknown consequences for electrophysiological studies, and the extent of sarcomere length, a well-recognized parameter of diastolic stretch, could not be determined in native slices during culture. Alternative assessment of sarcomere length in slices fixed inside the tissue chambers in diastole, i.e., under the specified preload, indicated that diastolic tension was within the physiologic range (Supplementary Fig. 4). In addition, biomechanical conditions can also be evaluated by the relationship of passive and active force developments. In the human heart, the contribution of diastolic to total systolic wall stress ranges from 11 to 23% in healthy individuals and in patients with volume overload, respectively[19]. These contractile parameters arise at 85–90% of the muscle length associated with maximum force development, corresponding to a sarcomere length of 1.85–2.0 μm in normal and diseased human trabecular fibres[21]. The conditions of biomimetic culture are well suited to approach these characteristics. A preload of 1 mN resulted in 83% of the muscle length required for maximum contractility (Fig. 3b, c), and the ratio of diastolic to systolic forces in culture declined from 23% at the beginning to 14% after 4 weeks of culture (Fig. 2d). Accordingly, a quite normal sarcomere length of 1.8 μm was confirmed in two tissue slices that had been fixed under standard stretch conditions (Supplementary Fig. 4).

To demonstrate the advantage of long-term exposure and repeated functional assessments, we tested the applicability of biomimetic culture for drug safety evaluation. Interference with cardiomyocyte repolarization frequently occurs as an undesired action of drugs, and may induce long-QT syndrome (LQTS) and associated arrhythmias[22]. The mechanisms of interaction can be diverse, and may affect a variety of ion channels, so that their evaluation demands new and more integrative approaches[23]. The antiprotozoal drug pentamidine is a prototypic example for high pro-arrhythmic risk conferred by interference with protein maturation, membrane transport and mRNA expression of the Kv11.1 channel (ether a-go-go-related gene, hERG)[24]. Due to the indirect and delayed mechanisms of action, hERG interference evades detection by protein interaction assays and acute electrophysiological testing. In contrast, our determination of refractory period in cultured human myocardium reflected the action potential duration in a therapeutically relevant tissue and documented the accumulating action of pentamidine over 2 weeks without analytical effort or intervention. The high efficacy of pentamidine after chronic exposure also indicated its interference with repolarising mechanisms other than hERG. Indeed, pentamidine and acute hERG blockade resulted in extensive action potential prolongation and early afterdepolarizations (Fig. 5d–g). Although the underlying mechanisms need to be resolved, such an observation would dictate a more severe risk estimation, particularly with regard to interactions with other hERG blocking drugs or with genetic LQTS predisposition. In the case of pentamidine, such precautions could have been therapeutically relevant since the sensitivity of standard assays of hERG interference (5.1–11.3 μM[24,25]) appears to underestimate the risk of arrhythmia at therapeutic mean plasma concentrations which are in the range of 0.1–0.5 μM[26].

In summary, the presented technology of biomimetic cultivation of myocardial tissue slices has the potential to promote research on human myocardium by permitting collaborative exchange of tissue samples and by implementing an automated procedure for the generation of multiple tissue cultures from one myocardial specimen. Long-term maintenance in tissue culture can provide a versatile source of human myocardium for many forms of acute experimentation. Furthermore, the demonstrated consistency of function and structure of cultivated myocardium will enable more realistic disease modelling, as well as assessment of the long-term outcomes of pharmacologic, cellular or genetic interventions.

## Methods

**Tissue acquisition**. Myocardial tissue specimens were obtained by the Clinic of Thoracic and Cardiovascular Surgery, Heart and Diabetes Center, Bad Oeynhausen, Germany and by the Clinic of Cardiac Surgery, University Hospital, Munich, Germany as $2 \times 2$ cm$^2$ transmural sections of left ventricular myocardium taken from failing hearts at the time of transplantation. In one instance, hypertrophic myocardium was utilized that had been resected for correction of subaortic outflow obstruction (Morrow procedure). This tissue did not display obvious differences to failing myocardium and was therefore included in this study. Patients provided informed consent to the scientific use of the explanted tissue. The study was approved by the institutional ethics boards of the clinical and the experimental study contributors (H.M., R.S., A.D.), and has been performed in accordance with the ethical standards laid down in the 1964 Declaration of Helsinki and its later amendments. Myocardial specimens were placed immediately in cold (4 °C) slicing buffer (136 mM NaCl, 5.4 mM KCl, 1 mM MgCl$_2$, 0.33 mM NaH$_2$PO$_4$, 10 mM glucose, 0.9 mM CaCl$_2$, 30 mM 2,3-butadione-2-monoxime, 5 mM HEPES, pH 7.4). Samples from the remote clinic were sent on ice by standard courier to the Walter-Brendel-Centre, Munich, resulting in an 18–32 h delay before slice preparation.

**Slice preparation and BMCC setup**. Tissue specimens were freed from endo-cardial trabecular layers and were trimmed to ~$8 \times 8$ mm$^2$ cross-sectional area. Tissue blocks were embedded in 4% low-melt agarose (dissolved in slicing buffer at 60 °C and applied to the tissue at 37 °C) and were mounted onto the cooled (4 °C) stage of a vibratome (VT1200S, Leica Biosystems, Germany). The tissues were cut in tangential orientation proceeding from the endo- to epicardial layers of the myocardium (1.5 mm vibration amplitude, 0.07 mm s$^{-1}$ feed rate, razor blade "Gillette Silver Blue"). A detailed description of the slicing procedure has recently been published[7,9]. Slices (300-μm thick) were glued (Histoacryl, B.Braun Melsungen AG, Germany) to small plastic triangles cut from 0.1-mm-thick polyester copier clear film (MGW5504, MGW Office Supplies, Germany) so that a $5 \times 5$ mm$^2$ muscle area was placed with aligned fibre orientation in between the fixation. Prior to the transfer to BMCCs, slices were stored for up to 1 h in cold slicing medium. Freshly prepared slices were attached with minimum preload to the elastic and the fixed holding wires of BMCCs and were submerged in 2.4 mL prewarmed (37 °C) culture medium. BMCCs were equilibrated under culture conditions for a 1 h period after which preload was adjusted to 1 mN, and stimulation was initiated. Preload was readjusted to 1 mN after 24 h of cultivation and was not further manipulated throughout the subsequent culture period. Individual steps of tissue slice preparation and mounting are illustrated as Supplementary Methods and in the Supplementary Movie 1.

**Culture conditions**. Complete culture platforms comprising eight BMCCs each, were placed in a standard incubator (37 °C, 5% CO$_2$, 20% O$_2$, 80% humidity), and were agitated continuously on a rocker plate (60 rpm, 15° tilt angle, Hi/Lo-Rocker, IBI Scientific, USA, equipped with brushless motor BLH230KC-30, Orientalmotor, Germany). Pacing was performed at 0.2 Hz with bipolar 50 mA pulses comprised of 1 ms charging and discharging pulses separated by a 1 ms interval. The unphysiological low beating rate was chosen in order to keep the demands of O$_2$ and nutrients at a minimum. Slices were cultured in Medium 199 supplemented with penicillin/streptomycin, insulin/transferrin/selenite and 2-mercaptoethanol (50 μM). Medium was exchanged in part (1.6 ml of 2.4 ml total volume in each BMCC) at 36–48 h intervals. For prolonged application, substances were dissolved at 1000-fold final concentrations in either DMSO (pentamidine) or 0.9% NaCl (denopamine) and were added to fresh medium with every medium exchange. Denopamine was chosen as an adrenergic agonist because of its reasonably selective efficacy at β$_1$ as opposed to β$_2$ and β$_3$ adrenoceptors[27]. In addition, denopamine lacks the catechol moiety that is prone to metabolic degradation.

**Design of biomimetic culture chambers (BMCCs)**. BMCCs were implemented either with manually reworked standard cell culture dishes (Falcon 353001), or with CNC-milled dishes made of polyoxymethylene or cyclic olefin copolymers (Fig. 1b, c). For elastic mounting of the muscle specimen, the chambers were equipped with a 0.25-mm-diameter steel wire (grade 1.4401, 18 mm long), the extended free end of which was labelled with a small magnet (NdFeB, 1 mm length, 0.5-mm diameter, HKCM engineering, Germany). Muscle preparations were fixed to the centre of this wire, and thereby were exposed to a spring constant of 75 mN/mm. The opposing end of the muscle was hooked to a steel wire (grade 1.441, 0.4 -mm diameter) that was led out of the chamber through a silicone seal, and that was connected at its other end to a manual linear drive (Fig. 1a). Deflection of the flexible steel wire was recorded by changes of the magnetic field that was taken up by a 3D-magnetic field sensor (FXOS8700, NXP Semiconductors, USA) placed in proximity to the magnet, but outside the culture chambers. The magnetic sensors were mounted to small electronic boards that were either integrated within the BMCCs (manual version, Fig. 1b), or reversibly

bolted to the BMCCs (CNC version, Fig. 1c). Electrical stimulation was provided by two $6 \times 8$ mm$^2$ graphite electrodes (CG1290, CGC Klein, Germany) that were connected by steel wires (grade 1.4401, 0.4-mm diameter) to the sensor boards, and were placed at 20 mm distance at both sides of the tissue (Fig. 1a, d).

**Stimulus generation and force monitoring**. Control of BMCCs was implemented with a FRDM-K22F development board (NXP Semiconductors, USA) featuring an ARM Cortex M4 microcontroller. This was programmed to collect magnetic field data from eight BMCCs via the integrated I$^2$C serial bus at 500 s$^{-1}$ sample rate, and to generate individual stimulation impulses that were distributed to each BMCC by a multiplexer (ADG1407, Analog Devices Inc., USA, Fig. 2a). Bipolar, current-controlled stimulation pulses were generated with an adjustable current source (LT1970, Linear Technology, USA). Microcontroller and stimulator were combined with eight BMCCs on a system board that was operated in a standard CO$_2$-incubator (Fig. 1f). Via USB, the microcontroller was connected to a PC that served as a recorder of the magnetic data and as a terminal for stimulation commands. In that way, all stimulation parameters (stimulus sequence and rate, pulse durations and currents) could be remotely controlled and manipulated by freely program-mable stimulation schedules (Fig. 1g). Real-time display of the contractility data and stimulation control were implemented with custom-written software utilizing the free library "Oscilloscope_DLL" (M. Bernstein, https://www.oscilloscope-lib.com). Recorded data were imported into and analysed by "LabChart Reader" software (AD Instruments, Australia).

**Characterization of tissue biomechanics**. Contractility of fresh and cultured myocardial slices was determined under isometric conditions in an organ bath, as previously described[7]. In brief, tissue was superfused with oxygenated KHS (37 °C, 4 mL min$^{-1}$), exposed to 1 mN preload, and to electrical stimulation (1 Hz, 3 ms pulse width, 1.5-fold stimulation threshold). Maximum twitch force was determined under optimum preload and isoprenaline stimulation (1 μM). Tonic contracture eventually occurred at the start of incubation and was quantified as the spontaneous increase of diastolic tension over 20 min. During biomimetic culture, preload and contractility were derived from the deflection of the BMCC spring wire, and were recorded continuously (Fig. 2b, c).

**Determination of refractory period**. Refractory period was regularly tested in BMCCs under undisturbed culture conditions. An automated stimulation protocol added a second pacing impulse at defined intervals after each regular stimulus. The intervals between both stimulations were decreased from 1000 to 250 ms in scheduled steps of 30 s duration. The longest interval that failed to induce two distinct contractions was regarded as equivalent to the refractory period (Fig. 5a).

**Histology**. Thin paraffin sections of tissue slices were stained with hematoxylin–eosin using standard techniques. Immunohistochemistry was done on whole-mount heart slices that had been fixed in 4% paraformaldehyde for at least 24 h. The tissues were equilibrated with a graded series of 4, 15 and 30% sucrose in PBS, and were permeabilized with 1% Triton X-100 overnight. After blocking (3% BSA in PBS, 12 h), samples were incubated sequentially with primary antibodies (anti-α-actinin, A7811, anti-connexin-43, C6219, anti-α-smooth muscle actin, A5228, all Sigma-Aldrich, anti-vimentin, AB92547, Abcam, anti-N-cadherin, #610921, BD-Biosciences, all 1:100), and secondary antibodies (anti-rabbit-Alexa488, A21441, anti-mouse-Alexa546, A11030, both ThermoFisher, 1:100, combined with DNA-stain 1 μM TO-PRO-3) for 1 day each. Washing steps between all incubations used citrate-buffer (150 mM NaCl, 15 mM Na$_3$-citrate, pH 7.2) supplemented with 1% BSA, 0.05% Triton X-100 and 3 mM NaN$_3$. Slices were mounted in VectaMount AQ (Vector Laboratories). Confocal microscopy was performed at the bioimaging core facility of the Biomedical Center using an inverted Leica SP8X WLL microscope.

**Transcriptome analysis**. Messenger RNA was sequenced in myocardial slices from three patients prior to and after 8, 14, 24 and 35 days of culture. Slices analysed at days 14 and 35 were from the same patient and tissue sample.

Strand-specific, polyA-enriched RNA was sequenced as described earlier[28]. Briefly, RNA was isolated from whole-cell lysates using the AllPrep RNA Kit (Qiagen) and RNA integrity number (RIN) determined with the Agilent 2100 BioAnalyzer (RNA 6000 Nano Kit, Agilent). For library preparation, 1 μg RNA was poly(A) selected, fragmented and reverse transcribed with the Elute, Prime, Fragment Mix (Illumina). End repair, A-tailing, adaptor ligation and library enrichment were performed as described in the Low Throughput protocol of the TruSeq stranded RNA Sample Prep Guide (Illumina). RNA libraries were assessed for quality and quantity with the Agilent 2100 BioAnalyzer and the Quant-iT PicoGreen dsDNA Assay Kit (Life Technologies). RNA libraries were sequenced as 100 bp paired-end runs on an Illumina HiSeq4000 platform. On average, we produced about 5.8 Gb of sequence per sample. The STAR aligner (v2.4.2a)[29] with modified parameter settings (--twopassMode = Basic) was used for split-read alignment against the human genome assembly hg19 (GRCh37) and UCSC knownGene annotation. To quantify the number of reads mapping to annotated genes, we used HTseq-count (v0.6.0)[30]. Differentially expressed genes were identified using the R Bioconductor package DESeq2 (v1.10.1)[31]. Enriched Gene Ontologies and pathways were investigated using the R Bioconductor packages goseq (v1.22.0)[32] and gage (v2.20.0)[33], respectively.

Culture-related transcriptional regulation of single genes or gene ontologies was expressed as the difference of log2-values of transcript numbers in the cultured and the uncultured tissue sample of the same patient. The time course of this parameter was approximated by linear regression to distinguish between early transcriptional responses and long-term alterations in tissue culture. The parameters of regression analysis yielded an extrapolation of expression levels at the start of cultivation (day 0-intercept), and of the temporal trend during 35 days of culture (slope multiplied by 35).

**Membrane potential recording**. Membrane potential was determined by intra-cellular recording from individual myocytes in tissue slices[7]. Sharp electrodes were drawn from GB100F-10 capillaries (Science Products), and backfilled with 2 M KCl (resistance 5–20 MOhm). Tissue slices were mounted at relaxed length in a heated organ bath and were superfused with HEPES-buffered Earle's-salt solution (pH 7.4, 4 mL min$^{-1}$, 37 °C). The electrode was advanced into the tissue until a stable resting potential was obtained. Field stimulation was performed with monopolar 1 ms pulses at the minimum effective strength. Potential was amplified with a BA-01X amplifier (npi electronic, Germany), and recorded with a PowerLab 4/20 system (AD Instruments, Australia). Substances were added consecutively into a recirculating 25 mL buffer volume.

**Morphometric analysis of myocyte structure**. Fresh and cultured cardiac tissue slices from three different hearts were fixed in 4% PFA and then prepared for immunofluorescence either directly (determination of cross section area) or after cryo-sectioning into slices of 50-μm thickness. Nuclei were stained with DAPI, α-actinin was stained with anti-α-actinin (A7811, Sigma-Aldrich) primary antibody and goat anti-mouse-AF488 secondary antibody (A21121, ThermoFisher) in concentrations 1:200 in PBS with 0.25% Triton X. Extracellular matrix and cell membranes were stained with 40 μg mL$^{-1}$ wheat germ agglutinin (WGA) con-jugated to AF647 (W32466, ThermoFisher). For imaging of cross-section areas, tissue slices were mounted free of compression in Fluoromount-G (#17984-25, Electron Microscopy Science, Hatfield, PA, USA)[34]. Otherwise, slices were mounted on a glass slide, embedded in Fluoromount-G and then covered by a coverslip. Three-dimensional microscopic image stacks were then recorded, using a Zeiss LSM780 confocal microscope. Image acquisition was carried out with 1280 × 1280 pixels per plane and approx. 200–300 planes per image. Voxel size was 0.1 × 0.1 × 0.2 μm, resulting in stack sizes of approx. 128 × 128 × 50 μm. We acquired at least three images from each sample at randomly selected regions.

Cross-section areas were determined by image processing and segmentation of cardiomyocytes using published methods[34,35]. Percentage of the extracellular matrix in the tissue was calculated from the volume fraction of segmented WGA signal and served as an indicator of fibrosis[36,37]. Sarcomere length was calculated from the Fourier transform of the α-actinin signal by determining the maximum in the power spectrum corresponding to spatial frequencies of 1/3 to 1 μm$^{-1}$, as recently described[38]. As a measure of the density of the transverse tubular system, we calculated the mean intracellular distance to the closest membrane after threshold-based segmentation of the WGA signal[39].

**Statistical information**. All data are given as mean values $+/-$ standard error of the means (SEM). Error bars indicate SEM. Statistical testing was performed using two-tailed $t$ tests, and one- or two-dimensional ANOVA, as appropriate. Statistical significance was accepted at an error level of $P<0.05$.

**Code availability**. Computer software developed for stimulation control and data recording, as well as technical details of culture chambers and electronic devices are available from the corresponding author upon reasonable request.

**Reporting Summary**. Further information on experimental design is available in the Nature Research Reporting Summary linked to this article.

## Data availability
The authors declare that all data supporting the findings of this study are available within the article and its Supplementary Information files or from the corre-sponding author. The Source data underlying Figs. 2a, 2d, 3a–d, 5b–c and Sup-plementary Figs. 1d, 2g–h, 3f and 4g are provided as Supplementary Data 2. Messenger RNA expression data are provided as Supplementary Data 1, and sequence readouts are available from the NCBI BioProject database under the accession number 496588. A reporting summary for this article is available as a Supplementary Information file.

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

## Acknowledgements

The authors want to thank Mr. F. Singer and Mr. F. Sojak for design and fabrication of various versions of the tissue incubation chambers. We also express our gratitude to Dr. E. Graf and Dr. P. Lichtner for performing transcriptome sequencing. The indispensable contribution of Mrs. D. Gerdes, Mrs. D. Baurichter, Mrs. K. Kämpf and Mrs. C. Stanasiuk with the collection of human myocardial tissue samples is gratefully acknowledged. Special thanks also go to Dr. J. Davis for thorough reading and linguistic revision of the manuscript. This study was supported by the German Center for Cardiovascular Research (grant 81×2600217 to H.M. and A.D.), and by the FöFoLe scholarship of the Ludwig-Maximilians-University Munich (C.F.).

## Author contributions

H.M., U.P. and A.D. designed the study. C.F., E.F. and A.D. developed the biomimetic incubation chambers and culture conditions. H.M., R.S. and R.T. acquired clinical data and tissue samples. C.F., E.F., E.V., K.L. and X.C.-E. performed slice preparations, tissue culture and real-time assessments. C.F., E.F. and B.H. established cold preservation and end-point biomechanics. E.V. and B.H. evaluated chronic and electrophysiological effects of pentamidine. K.L. and C.S. analysed tissues by conventional and immunostaining. T.S. and R.T performed quantitative morphometry. T.Sch. and X.C.-E. analysed transcriptome data. C.F., T.S., T.Sch., U.P. and A.D. interpreted data and wrote the paper.

## Additional information

**Competing interests:** A patent application has been filed by A.D. covering the integration of a magnetic force sensor into a cell culture device (USN 15/781,454). Other author declare no competing interests.

**Journal Peer Review Information:** *Nature Communications* thanks the anonymous reviewers for their contributions to the peer review of this work. Peer Reviewer Reports are available.

