## [Peer Review File · Nature Communications]

Reviewers' comments:

Reviewer #1 (Remarks to the Author):

The manuscript describes an improved method to preserve structural and functional integrity of **human adult** cardiac tissue slices (also called: organotypic culture).

The most **impressing result** is the long-term viability of the tissue und the described conditions.

Many groups have developed culture conditions to improve longevity of **adult** cardiac tissue slices before. While it is relatively easy to maintain embryonic or neonatal cardiac tissue viable for several days, adult cardiac tissue shows a rapid loss of viability and dedifferentiation, precluding chronic experiments. Therefore, many experiments with **adult** cardiac tissue slices usually should be performed within hours or at least on the same day to be relevant.

In their manuscript, Fischer et al. demonstrate structural and functional integrity for “up to 4 months”. This is far more than most groups achieved before. It would be even long enough to perform real **chronic** experiments. Experiments that analyse remodelling or the effects of adenoviral-mediated gene transfer should be possible under these circumstances.

Specific Comments for Authors:

Title

The term “biomimetic system” is very imprecise. Today, many things are termed “biomimetic” (<https://en.wikipedia.org/wiki/Biomimetics>). Since the whole work is about tissue slices, I would suggest using this term already in the title.

Abstract

Abstract – Line 32-33:

“In biomimetic culture, they developed a stable state of contractility that was monitored for up to 4 months or 2000000 beats in vitro.”

This would correspond to a beating rate of 11-12 bpm. This is far below the typical adult beating rate and not physiological. The authors are asked to comment this.

Introduction (though there is no headline like this)

The authors nicely describe the numerous models that have been developed during the last years. They describe that tissue culture techniques have been modified and improved, tissue slices have become very popular, they report that mechanical stimulation and electrical stimulation have already been used to increase the longevity and viability of the preparations.

However, the fail to elaborate what part of their model is the most important development, the crucial part, why their model is superior to others and why their approach is worth while to be published and why others should switch to their model or how they might improve their current models.

The authors are asked to provide this information to the reader (“what is new”).

Results:

Line 96 / Figure 1

“Myocardial biomechanics and excitation are provided by a versatile 8-channel biomimetic culture setup.”

To my mind **lines 94-126** belongs to “Methods” and not to “Results”

Figure 1 b, c, d and f: A scale bar should be added.

Since the main purpose of manuscripts like this is to provide other investigators with a model that they can reproduce a very detailed description of the model is mandatory. If the number of words is limited by the journal, a supplement could be useful.

If I would try to reproduce the model, the following information would be helpful:

Line 101:

“slices were glued to thin (0.1 mm) plastic triangles”

What is the material of plastic triangles?

Lines 104 ff:

“A linear relationship between shortening and contraction force was implemented by fixation of one end of the muscle to a steel spring wire whose elastic constant (75 mN/mm) was chosen so as to generate normal systolic left ventricular wall tension (15 kN/m²) at about 6 % shortening of a typical myocardial specimen of 5x5x0.3 mm³ dimensions.”

A video would be a very useful supplement.

For a better understanding of the optimal tension for a longtime survival the authors should provide information that can be transferred to other models:

The sarcomere length should be measured (see e.g. Holubarsch C. et al. Circulation 1996) because it is more relevant than the estimation of the wall tension (the dimension of the tissue slice that is vital might be difficult to obtain).

The authors studies end-stage human failing myocardium. The optimal sarcomere length is probably more relevant than the numbers that are provided.

Could the authors measure and/or provide the corresponding sarcomere length during culture.

It remains unclear whether the developed tension was tested and/or retested during the experiments and how often. Could the authors please provide this information in the text.

Line 112 ff:

“a 3D magnetic field sensor that was placed at a distance generating a maximum of 1.2 mT magnetic flux at right angle to the sensor surface. [...]”

Hopefully, I did not miss this in the Supplement:

The authors should specify the stimulation rate and the reason for choosing it.

“Explanted failing myocardium maintains viability and function after up to 32 h of cooled transport” (lines 127-143)

Line 130 ff:

Did the authors test right ventricular or atrial slices as well?

Did the authors observe spontaneous contractions?

“Adult failing myocardium attains a stable state of continuous activity during long-term biomimetic cultivation” (lines 145-180)

Figure 2e

Why did the authors use denopamine instead of more frequently used and more selective beta1-receptor agonists?

Figure 2f

The scale for the developed force is missing, the x-axis can only be derived from the grey areas.

Figure 3

Line 747:

“No differences between fresh and cultured tissues could be confirmed (2-way ANOVA).”

This seems to be hard to believe when looking at the figures.

The authors might explain the obvious discrepancy.

“Biomimetic cultivation preserves tissue structure and induces distinct alterations in gene expression.” (lines 181-225)

Given the high quality and detailed information provided by light microscopy, ultrastructural data would have been nice but are not necessary any more.

“Repeated assessment of refractory period reveals repolarization impairment induced by hERG-channel (Kv11.1, KCNH2) interference.” (lines 227-244)

Figure 5d-g.

The authors should provide a scaling for both the x- and y-axes

Discussion:General comments:

The authors have developed the model further and discuss the most important aspects of this improvement; they should however, point out the most important change they made, the most important contribution they made as compared to previous research groups. And, I think there should be a section that points out the limitations.

Another very important point would have been to discuss the temporal limitations:

When experiments are done with this experimental set-up:

What is the time frame for physiologically relevant studies?

The authors may comment on this topic more precisely.

Methods:General comment:Reproducibility of the model:

The authors describe an interesting further development of already existing tissue culture models and prove its relevance. For this reason, one of the most important topics is, that other researchers can reproduce the model. Therefore, a very precise and very detailed description of the model is mandatory. Although they might quote their previous work, other researchers should be able to use this technique with the help of this paper alone.

Supplemental videos would be helpful.

Reviewer #2 (Remarks to the Author):

Authors show that they can keep human heart slices healthy and functional for weeks, if not months, under optimal stretching conditions and with regular electrical stimulation and medium exchanges. The physiology done to confirm the viability and robustness of the specimens is well-executed, and the results are convincing. Moreover, they go on to show clinically-relevant utility in demonstrating measurable responses (action potential duration changes) to drugs known to perturb the QT interval. Thus, it is likely that preps could be used to screen novel compounds for undesirable QT prolonging effects, an important go/no go bottleneck in global drug development efforts. Such screening is now being done on selective HERG channel-expressing kidney cells, but there may be false positives which could be more trenchantly studied in a human myocardial context, potentially rescuing otherwise-viable therapeutic candidates. My only suggestion is to add a table showing the clinical characteristics of the patients from whom the tissues were derived (i.e., are these sick hearts or not, ages, sex, etc.), and also more factual information on success rates.

Responses to Referees' comments on manuscript "Long-term cultivation of continuously beating human myocardium in a novel multi-channel biomimetic system"

Responses to comments of Referee #1:

Thank you for your detailed and very constructive critique. Below, we list the modifications of the manuscript taken to address your specific comments:

General Comments for Author:

*The manuscript describes an improved method to preserve structural and functional integrity of **human adult** cardiac tissue slices (also called: organotypic culture).*

*The most **impressing result** is the long-term viability of the tissue und the described conditions.*

*Many groups have developed culture conditions to improve longevity of **adult** cardiac tissue slices before. While it is relatively easy to maintain embryonic or neonatal cardiac tissue viable for several days, adult cardiac tissue shows a rapid loss of viability and dedifferentiation, precluding chronic experiments. Therefore, many experiments with **adult** cardiac tissue slices usually should be performed within hours or at least on the same day to be relevant.*

*In their manuscript, Fischer et al. demonstrate structural and functional integrity for “up to 4 months”. This is far more than most groups achieved before. It would be even long enough to perform real **chronic** experiments. Experiments that analyse remodelling or the effects of adenoviral-mediated gene transfer should be possible under these circumstances.*

Specific Comments for Authors:

Title

The term “biomimetic system” is very imprecise. Today, many things are termed “biomimetic” (<https://en.wikipedia.org/wiki/Biomimetics>). Since the whole work is about tissue slices, I would suggest using this term already in the title.

It is difficult to express the essential information of a study with the 15 words allotted to the Title. The attribute "biomimetic" in the most common context is used to indicate the imitation of a specific biological property or structure by artificial means. However, in the context of bioengineering, the term is usually expanded to the simulation of more complex and less defined natural conditions with the aim to induce a physiological behaviour *in vitro*, e.g. in:

Rapid 3D bioprinting of decellularized extracellular matrix with regionally varied mechanical properties and **biomimetic** microarchitecture. Ma X, Yu C, Wang P, Xu W, Wan X, Lai CSE, Liu J, Koroleva-Maharajh A, Chen S. Biomaterials. 2018 Sep 18.

The description of our technical system as "biomimetic" was chosen to indicate that it exactly targets the same aim. We expected that the context of "biomimetic system" would clearly indicate the use of a technical approach instead of a biochemical or biological one. That kind of meaning is quite common for the description of cultivation conditions, as in:

Curvature facilitates podocyte culture in a **biomimetic** platform. Korolj A, Laschinger C, James C, Hu E, Velikonja C, Smith N, Gu I, Ahadian S, Willette R, Radisic M, Zhang B. Lab Chip. 2018 Sep 28

We agree that the Title subliminally suggests that the culture technique might be applicable to "human myocardium" in general. Indeed, we expect that the system will be usable for the cultivation of trabecular or papillary muscle strips, or human engineered heart tissues as well. On the other hand, attribution of general properties like "human" or "continuously beating" to a very specific item, like "myocardial tissue slice" would rather inappropriately be referred to the "slice" property rather than to that of the myocardial tissue, as in:

"Long-term cultivation of continuously beating human myocardial tissue slices in a novel multi-channel biomimetic system"

Under these considerations, we suggest the following modification of the Title:

"Preservation of function and structure of precision-cut human myocardium under continuous electromechanical stimulation *in vitro*"

However, as this Title would not reflect the high stability of the cultured tissue, nor the essential impact of a newly developed technical system, we would rather prefer the original Title. We would suggest to leave the choice of the Title to the discretion of the Editors.

Abstract

Abstract – Line 32-33:

"In biomimetic culture, they developed a stable state of contractility that was monitored for up to 4 months or 2000000 beats in vitro."

This would correspond to a beating rate of 11-12 bpm. This is far below the typical adult beating rate and not physiological. The authors are asked to comment this.

Continuous stimulation was done at a rate of 12 bpm (0.2 Hz, lines 168, 202, 438). The slow stimulation rate was chosen in order to keep the demands of O₂ and nutrients at a minimum. Indeed, it seemed to be critical to accommodate O₂ consumption at atmospheric O₂ partial pressure. Similarly, medium exchange intervals of not shorter than 2 days were desired. Since the low stimulation rate was sufficient to maintain contractility and excitability, we defined this as a standard condition. Most recent observations indicate that a beating rate of 30 bpm will be tolerated without impairment of long-term stability, and also 60 bpm might be achievable. However, the presented data have been acquired

with 12 bpm stimulation exclusively, and the experience with higher rates still has to be extended.

-> The considerations that favour a slow stimulation rate are now given in the "Methods" section (lines 438-440), its implications are discussed as a possible limitation of the model (lines 339-344)

Introduction (though there is no headline like this)

The authors nicely describe the numerous models that have been developed during the last years. They describe that tissue culture techniques have been modified and improved, tissue slices have become very popular, they report that mechanical stimulation and electrical stimulation have already been used to increase the longevity and viability of the preparations.

However, the fail to elaborate what part of their model is the most important development, the crucial part, why their model is superior to others and why their approach is worth while to be published and why others should switch to their model or how they might improve their current models.

The authors are asked to provide this information to the reader ("what is new").

As yet, preservation of contractility has not been possible in tissue slices, and has been achieved only in one earlier study reporting cultivation of trabecular myocardium for as long as 6 days (Ref. 13). In this case, a complex incubation chamber supported culture of only one sample at a time. Essentially, maintenance of contractility in tissue culture has only become feasible with our novel system. The indicated earlier incubator enforced isometric contraction, and applied unipolar stimulation pulses and 95 % fractional O₂ concentration. Accordingly, one can speculate that avoidance of systolic overload, electrochemical reactions, and/or oxygen radicals might represent the major advances of the new technology.

-> Innovations of our technique in these respects are now listed in the Discussion. We summarized these aspects as "integrated biomimetic" approach to indicate that their individual significance for the success of cultivation remains elusive (lines 278-292).

Results:

Line 96 / Figure 1

"Myocardial biomechanics and excitation are provided by a versatile 8-channel biomimetic culture setup."

To my mind lines 94-126 belongs to "Methods" and not to "Results"

For its major part, the study is about the development and properties of a new tissue cultivation method. So it seemed appropriate to describe technical details of the method in the "Methods" section, and to report the achievements (force distribution, preload, compliance, sensor sensitivity, electrode stability) as "Results". Some of the contents cannot clearly be assigned to either group. However, some description of the cultivation setup in the "Results" section is required, because schematic drawings and pictures of the technology need to be presented at the beginning of the article for the sake of readability.

-> We hope that these considerations justify our suggestion to make no changes.

Figure 1 b, c, d and f: A scale bar should be added.

-> Scale bars have been added to Figure 1b,c,d,f

Since the main purpose of manuscripts like this is to provide other investigators with a model that they can reproduce a very detailed description of the model is mandatory. If the number of words is limited by the journal, a supplement could be useful.

-> Details of slice preparation and biomimetic culture conditions are now more exactly described in an extended version of the "Methods" section (lines 412-448).

-> The procedure of tissue preparation, slice generation, and setup of a BMCC is now illustrated by a step-by-step picture sequence added under "Supplementary Methods".

-> The handling of slices, attachment of triangles, and slice mounting into a BMCC is now demonstrated in a supplementary video.

-> Technical details such as 3D-drawings of incubation chambers, circuit diagrams, and printed circuit board layouts will be provided on request.

If I would try to reproduce the model, the following information would be helpful:

Line 101:

"slices were glued to thin (0.1 mm) plastic triangles"

What is the material of plastic triangles?

-> The material is now specified in the "Methods" section (lines 422-423)

Lines 104 ff:

“A linear relationship between shortening and contraction force was implemented by fixation of one end of the muscle to a steel spring wire whose elastic constant (75 mN/mm) was chosen so as to generate normal systolic left ventricular wall tension (15 kN/m²) at about 6 % shortening of a typical myocardial specimen of 5x5x0.3 mm³ dimensions.”

A video would be a very useful supplement.

-> The procedure of tissue preparation, slice generation, and setup of a BMCC is now illustrated by a step-by-step picture sequence added under "Supplementary Methods".

-> The handling of slices, attachment of triangles, and slice mounting into a BMCC is now demonstrated in a supplementary video.

For a better understanding of the optimal tension for a longtime survival the authors should provide information that can be transferred to other models:

The sarcomere length should be measured (see e.g. Holubarsch C. et al. Circulation 1996) because it is more relevant than the estimation of the wall tension (the dimension of the tissue slice that is vital might be difficult to obtain).

The authors studies end-stage human failing myocardium. The optimal sarcomere length is probably more relevant than the numbers that are provided.

Could the authors measure and/or provide the corresponding sarcomere length during culture.

We agree that sarcomere length would be a valuable parameter for improved control of diastolic load. We regret that we have not been able to determine sarcomere length under cultivation conditions by laser diffraction. Either our optical system or the heterogeneity of fibre orientation in ventricular (as opposed to trabecular) heart muscle prevented this analysis. As a substitute, we determined sarcomere length in unloaded tissues by imaging and Fourier transformation, and found that under our conditions of preload adjustment, sarcomere length did not change during culture (Supplementary Figure 1).

We are also aware of the limitation that the cross sectional area of vital myocardium was not available for the calculation of wall stress during culture. To approach physiological conditions as close as possible, we chose preload so as to establish a physiological fraction of total systolic tension. In human heart, this ratio ranges from **11% to 23%** in healthy individuals and in patients with volume overload, respectively (Grossman W, Jones D, McLaurin LP.: Wall stress and patterns of hypertrophy in the human left ventricle. J Clin Invest. 1975;56(1):56-64). This mechanical performance is generated at **85-90 %** of the muscle length associated with maximum force development, and corresponds to a sarcomere length of 1.85-2.0 μm in normal and diseased human trabecular fibres (Vahl CF, et al.: Myocardial length-force relationship in end stage dilated cardiomyopathy and normal human myocardium: analysis of intact and skinned left ventricular trabeculae obtained during 11 heart transplantations. Basic Res Cardiol. 1997;92(4):261-70).

Under our conditions of biomimetic culture, the chosen preload of 1 mN resulted in **83%** of muscle length required for maximum contractility (derived from Fig. 3bc), and the ratio

of diastolic to systolic forces during culture declined from **23%** at the beginning **to 14%** after 4 weeks of culture (derived from Fig. 2d). As such, the established conditions are quite comparable to those of the beating heart, and it may be expected that this would also apply to sarcomeric distension.

This assumption could be confirmed in two recent myocardial samples that had been fixed within the culture dishes after 1 and 3 weeks of cultivation. Morphometric analysis demonstrated a mean and maximum sarcomere length of 1.8 μm and 2.0 μm , respectively, in each of them (Supplementary Fig. 4). Considering that sarcomere length is slightly underestimated with this method, due to cells that are not exactly parallel to the imaging plane, we suggest that our chosen preload resulted in sarcomere lengths in the optimal range of approx. 1.9 to 2.0 μm . Accordingly, our approach of defining preload at the initiation of biomimetic culture seems to be well suited to meet the conditions of natural human myocardium.

-> The procedure for adjustment of preload is now more exactly described under "Methods" (lines 425-430).

-> The consideration of sarcomere length and the uncertainty about its magnitude during culture have now been described as a possible limitation of the study (lines 343-358).

-> The sarcomere length of muscle fixed under culture conditions has been added (Supplementary Fig. 4).

It remains unclear whether the developed tension was tested and/or retested during the experiments and how often. Could the authors please provide this information in the text.

Developed tension was measured continuously over the whole cultivation periods at 500 samples s^{-1} . This is exemplified in Fig. 2bc, and described in the "Results" and "Methods" sections (lines 125 and 474, respectively). Amplitude and minima of developed force were presented as twitch force and diastolic preload in Fig. 2d. Wall stress was estimated on the basis of calculated cross section area.

Line 112 ff:

"a 3D magnetic field sensor that was placed at a distance generating a maximum of 1.2 mT magnetic flux at right angle to the sensor surface. [...]"

Hopefully, I did not miss this in the Supplement:

The authors should specify the stimulation rate and the reason for choosing it.

Continuous stimulation was done at a rate of 12 bpm (0.2 Hz, lines 168, 202, 438). The slow stimulation rate was chosen in order to keep the demands of O_2 and nutrients at a minimum. Indeed, it seemed to be critical to accommodate O_2 consumption at atmospheric O_2 partial pressure. Similarly, medium exchange intervals of not shorter than

2 days were desired. Since the low stimulation rate was sufficient to maintain contractility and excitability, we defined this as a standard condition.

-> The considerations that favour a slow stimulation rate are now given in the "Methods" section (lines 438-440), its implications are discussed as a possible limitation of the model (lines 339-344)

“Explanted failing myocardium maintains viability and function after up to 32 h of cooled transport” (lines 127-143)

Line 130 ff:

Did the authors test right ventricular or atrial slices as well?

Did the authors observe spontaneous contractions?

In order to maximize the number of homogenous tissue cultures, only left ventricular myocardium has been studied so far. Single spontaneous contractions occasionally occurred during warming up at the initiation of culture (first 60 min). Apart from that, triggered excitation was observed exclusively.

-> Information on spontaneous contractions has now been added to the "Results" section (lines 163-164)

“Adult failing myocardium attains a stable state of continuous activity during long-term biomimetic cultivation” (lines 145-180)

Figure 2e

Why did the authors use denopamine instead of more frequently used and more selective beta1-receptor agonists?

In contrast to adrenoceptor antagonists which are available with extensive β_1 selectivity, the choice of β_1 ligands with agonistic properties is limited. Denopamine has been identified in a large screen as the only agonist with pronounced selectivity in efficacy at β_1 vs. β_2 and β_3 receptors (Baker JG: The selectivity of beta-adrenoceptor agonists at human beta1-, beta2- and beta3-adrenoceptors. Br J Pharmacol. 2010;160(5):1048-61). In addition, denopamine lacks the catechol moiety that renders other agonists prone to oxidation and degradation. We could confirm that the activity of denopamine persisted throughout the time intervals of medium exchange (up to 48 h).

-> A reference to the selectivity and stability of denopamine has now been included into the "Methods" paragraph (lines 446-448).

Figure 2f

The scale for the developed force is missing, the x-axis can only be derived from the grey areas.

-> Scale bars have been added to Figure 2f.

Figure 3

Line 747:

“No differences between fresh and cultured tissues could be confirmed (2-way ANOVA).”

This seems to be hard to believe when looking at the figures.

The authors might explain the obvious discrepancy.

Indeed, prominent differences in mechanical properties of fresh and cultured myocardium have been observed, and accordingly, these have been addressed as results. The failure to verify these alterations statistically, reflected the high variation of samples properties, but was also owed to the inclusion of one experiment with an incomplete data set in the control group. After omission of this experiment, 2-way ANOVA confirmed suppression of contractility in cultured slices (although this could not be assigned to high stretch conditions specifically), and enhancement of force development with bradycardia. As such, description of these alterations in the "Results" section can remain unaltered, but their statistical significance needs no longer to be questioned.

-> Fig. 3 and its legend have been corrected.

-> Summaries of statistical testing have been added to the Source Data file.

“Biomimetic cultivation preserves tissue structure and induces distinct alterations in gene expression.” (lines 181-225)

Given the high quality and detailed information provided by light microscopy, ultrastructural data would have been nice but are not necessary any more.

Thank you for your appreciation.

“Repeated assessment of refractory period reveals repolarization impairment induced by hERG-channel (Kv11.1, KCNH2) interference.” (lines 227-244)

Figure 5d-g.

The authors should provide a scaling for both the x- and y-axes

-> Scale bars have been added to Figure 5d-g.

Discussion:

General comments:

The authors have developed the model further and discuss the most important aspects of this improvement; they should however, point out the most important change they made, the most important contribution they made as compared to previous research groups. And, I think there should be a section that points out the limitations.

-> Innovations of our technique are now discussed in more detail (lines 278-292).

-> The discussion of possible applications of the model has been extended to also cover its most obvious limitations (lines 328-358)

*Another very important point would have been to discuss the temporal limitations:
When experiments are done with this experimental set-up:
What is the time frame for physiologically relevant studies?
The authors may comment on this topic more precisely.*

The phases of contractile recovery and of stable performance may both reveal interesting information on cardiac physiology and pathophysiology. The stable phase may be more suitable for drug testing since it enables long-term observations.

-> Considerations about the best conditions and time-frames for physiological and pharmacological studies have been integrated into the discussion of possible applications and limitations of the model (lines 332-338).

Methods:

General comment:

Reproducibility of the model:

The authors describe an interesting further development of already existing tissue culture models and prove its relevance. For this reason, one of the most important topics is, that other researchers can reproduce the model. Therefore, a very precise and very detailed description of the model is mandatory. Although they might quote their previous work, other researchers should be able to use this technique with the help of this paper alone.

Supplemental videos would be helpful.

-> Details of slice preparation and biomimetic culture conditions are now more exactly described in an extended version of the "Methods" section (lines 412-448).

-> The procedure of tissue preparation, slice generation, and setup of a BMCC is now illustrated by a step-by-step picture sequence added under "Supplementary Methods".

-> The handling of slices, attachment of triangles, and slice mounting into a BMCC is now demonstrated in a supplementary video.

We hope that the extension of methodological details and the addition of visual material demonstrating the procedures of slice generation and culture setup will enable the establishment of the culture technique in as many research labs as possible. As already stated, any technical support for the reproduction or advancement of the system will be provided on request.

Responses to Referees' comments on manuscript "Long-term cultivation of continuously beating human myocardium in a novel multi-channel biomimetic system"

Responses to comments of Referee #2:

Thank you for sharing our fascination for the investigation of human myocardium *in vitro*. Below, we list the modifications of the manuscript taken to address your specific comments:

Authors show that they can keep human heart slices healthy and functional for weeks, if not months, under optimal stretching conditions and with regular electrical stimulation and medium exchanges. The physiology done to confirm the viability and robustness of the specimens is well-executed, and the results are convincing. Moreover, they go on to show clinically-relevant utility in demonstrating measurable responses (action potential duration changes) to drugs known to perturb the QT interval. Thus, it is likely that preps could be used to screen novel compounds for undesirable QT prolonging effects, an important go/no go bottleneck in global drug development efforts. Such screening is now being done on selective HERG channel-expressing kidney cells, but there may be false positives which could be more trenchantly studied in a human myocardial context, potentially rescuing otherwise-viable therapeutic candidates.

My only suggestion is to add a table showing the clinical characteristics of the patients from whom the tissues were derived (i.e., are these sick hearts or not, ages, sex, etc.), and also more factual information on success rates.

A description of patient characteristics is now provided with the Supplementary Table 1. We realized that in addition to the 18 samples of failing explanted heart, we also used one piece of hypertrophic myocardium, taken at surgical correction of subvalvular outflow obstruction (Morrow procedure). There was no obvious difference in the functional features and culture performance of this tissue, so we opted to keep it in the data pool because it gives a hint to the possibility to also investigate such tissues in long-term culture.

Information on reproducibility and success rates of the cultivation procedure has been extended with a more detailed description in the "Results" section (lines 164-175). The success of cultivation has now been listed for each tissue specimen, and has been associated with the medical history of the respective patient (Supplementary Table 1).

REVIEWERS' COMMENTS:

Reviewer #1 (Remarks to the Author):

Dear Authors,

With respect to my comment on the Title:

I agree, that "long-term" should be included in the title and was not aware of the limitation to 15 words for the title. I agree, It should b the Editor's decision, whether he would allow 16 words.

With respect to my other comments:

I think that your modifications, new supplementary figures and videos have improved the manuscript significantly. These changes will allow other researchers to reproduce your model.

Responses to Referees' comments on the manuscript

"Long-term functional and structural preservation of precision-cut human myocardium under continuous electromechanical stimulation *in vitro*"

Responses to comments of Referee #1:

Dear Authors,

With respect to my comment on the Title:

I agree, that "long-term" should be included in the title and was not aware of the limitation to 15 words for the title. I agree, It should b the Editor's decision, whether he would allow 16 words.

⇒ The Title has been modified according to the Editor's suggestion.

With respect to my other comments:

I think that your modifications, new supplementary figures and videos have improved the manuscript significantly. These changes will allow other researchers to reproduce your model.

⇒ Thank you for thorough evaluation of the study and for your valuable advice.